# Intelligent Control in Embodied Robotics: Enhancing Human-Robot Interaction through Adaptive Control Techniques

## Abstract

Current embodied intelligence models often lack the ability to adjust control methods dynamically in response to human intentions, limiting their effectiveness in real-world interactions. This paper proposes a novel framework that enables robots to dynamically adapt their control parameters by integrating large language models (LLMs) with intelligent controllers. Our approach simulates human-robot interactions and generates synthetic training data, allowing robots to better understand and respond to diverse human needs. We validate the framework using two commonly used control techniques and demonstrate that it can effectively adjust control methods, such as Proportional-Integral-Derivative (PID) and Nonlinear Model Predictive Control (NMPC), based on real-time human feedback. Experimental results show that our model enhances adaptability and responsiveness in human-robot interaction. This work advances embodied intelligence by introducing an adaptive control framework and providing a scalable method for data generation, which together enable more intuitive and effective robot behaviors.

## 1 Introduction

Recent advancements in large language models (LLMs) have profoundly impacted various sectors within the field of artificial intelligence (AI) Raiaan et al. (2024). These models, capable of processing and generating human-like text, have paved the way for integrating AI with physical systems, particularly in robotics. The amalgamation of LLMs with robotics aims to leverage their natural language processing capabilities to enhance the functionality and adaptability of robots in the physical world Pfeifer & Iida (2004); Haugeland (1989); Pfeifer & Bongard (2006). This convergence heralds a new era wherein the sophisticated understanding and generation of human language by LLMs are employed to create intelligent agents capable of interacting with and manipulating their physical environment. These intelligent agents, operating within the physical realm, are often referred to as embodied AI Turing (1950).

Embodied AI represents a significant shift from traditional, disembodied AI that exists solely in cyberspace to a form where AI extends its capabilities into the tangible world. This transition not only bridges the gap between digital intelligence and physical action but also broadens the scope of AI applications, encompassing sim-to-real robotic control, embodied agents, and embodied interaction Liu et al. (2024). Each of these areas presents unique challenges and opportunities for enhancing human-robot interaction and improving the overall effectiveness of robotic systems.

### 1.1 Research Motivation and Objectives

Achieving embodied intelligence requires a deep understanding of both the physical dynamics of the environment and the subtleties of human communication. For instance, a robot capable of embodied intelligence must be able to recognize and interpret human gestures, facial expressions, and vocal tones, and then respond appropriately. This level of interaction necessitates sophisticated sensory and processing capabilities, as well as advanced algorithms for real-time decision-making and action execution Jin et al. (2024); Li et al. (2024); Newbury et al. (2023); Varley et al. (2017); Fang et al. (2020); Xu et al. (2023); Shridhar et al. (2022); Shen et al. (2023).

Robot control is a multi-layered process that integrates high-level decision-making with low-level actuation to achieve autonomous and efficient operation in dynamic environments. The standard hierarchical model comprises three layers: decision, planning, and execution. Current state-of-the-art models typically facilitate human-robot interaction at the decision or planning layers. In this paper, we propose a novel model that enables human-robot interaction at lower layers Figure 1, specifically at the execution layer, thus Enhancing Robots' Empathy(ERE).

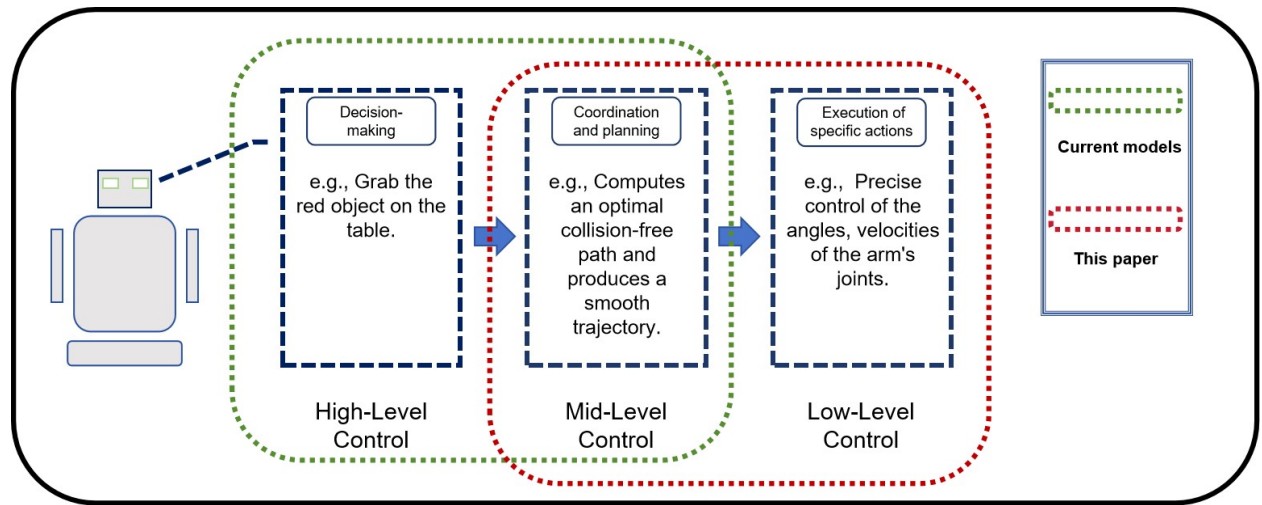

Figure 1: Robot control levels.

This empathetic approach to robotics can significantly improve the intuitiveness and effectiveness of human-robot interactions, particularly in fields such as healthcare, education, and assistive technologies.

## 1.2 CONTRIBUTIONS

This paper makes the following contributions:

- We introduce a novel approach that enhances robots' empathy and response to human intentions, extending these capabilities to the control level in robotics.

- We propose a data generation framework that uses large language models to simulate scenarios and inter-actions, creating synthetic data that closely mimics real-world conditions. This framework addresses the challenge of data scarcity and improves the robustness and generalizability of AI models.

- We demonstrate the effectiveness of our approach through extensive experiments, showing that robots can alter their control styles based on user requirements and the conditions of their environment.

The remainder of the paper is organized as follows: Section 2 introduces related work in the field of embodied intelligence. Section 3 details the methodology of Enhancing Robots' Empathy (ERE), including implementation methods and data generation techniques. Section 4 presents experimental results. The paper concludes with a summary of findings and contributions and discusses potential improvements and future research directions in Section 5.

## 2 RELATED WORK

The integration of perceptual data, cognitive processing, and action execution in robotics, often referred to as embodied intelligence, has been a focus of recent research efforts. This section discusses significant advancements in this field, emphasizing multimodal integration, end-to-end learning, and action generation, which are critical for developing adaptive, context-aware robotic systems.

**Advancements in embodied intelligence.** The landscape of embodied intelligence has been shaped by several influential models that leverage multimodal data to enhance robotic capabilities. Jang et al. Jang et al. (2021) introduced BC-Z, a model utilizing robotic imitation learning for zero-shot task generalization. Unlike conventional approaches requiring extensive task-specific training, BC-Z employs behavioral cloning, enabling robots to adapt to new tasks without prior explicit training, marking a significant shift towards more flexible robotic systems.

Similarly, Ahn et al. Ahn et al. (2022) proposed a model that integrates natural language understanding directly into robotic affordances, allowing robots to perform tasks based on verbal instructions. This model does not fully exploit multimodal capabilities but significantly enhances the interaction between verbal commands and robotic actions by grounding language in physical context Ahn et al. (2022).

Furthermore, the RT-2 model by Brohan et al. Brohan et al. (2023) exemplifies the integration of vision and language for robotic control. Utilizing web-sourced knowledge, this end-to-end multimodal model can generate precise action commands from visual and linguistic inputs, effectively bridging the gap between digital information processing and physical task execution Brohan et al. (2023).

**Innovative multimodal and end-to-end models.** Continuing with the theme of multimodal interaction, Driess et al. Driess et al. (2023) developed PaLM-E, an embodied multimodal language model that combines visual, linguistic, and embodied data. This model is particularly adept at understanding complex commands and executing them within a physical environment, demonstrating significant improvements in contextual understanding and robotic responsiveness Driess et al. (2023).

In a similar vein, VoxPoser by Huang et al. Huang et al. (2023) integrates 3D value maps with language models to facilitate complex robotic manipulations. This system not only enhances the spatial reasoning capabilities of robots but also allows for precise manipulation based on language-driven planning Huang et al. (2023).

**Expanding the scope of robotic applications.** The scope of embodied intelligence is further expanded by models like the foundation model-based system for open vocabulary task planning by Obinata et al. Obinata et al. (2023), which enhances service robots' ability to understand and execute a wide range of tasks based on open-ended commands. This flexibility is crucial for adapting to diverse service environments Obinata et al. (2023).

Moreover, the integration of ChatGPT as a vehicle co-pilot by Wang et al. Wang et al. (2023) explores the potential of conversational AI in improving automotive safety and interaction, highlighting the broader applicability of AI technologies in daily life and specialized settings Wang et al. (2023).

**Large language models used in reward functions.** There is also a work that Yu et al. (2023) primarily focuses on high-level action control by leveraging large language models to translate natural language instructions into reward functions, enabling robots to understand human language and perform tasks. Additionally, another work applies large language models directly to the design of reward functions, improving the reinforcement learning performance of robots Ma et al. (2024). **Large language models as optimizers.** Large language models have also been explored for iteratively solving classical problems. In this paper, they address challenges such as path planning through iterative use of large language models, revealing their potential in the direction of iterative optimization Yang et al. (2024).

The advancements in embodied intelligence reflect a broader trend towards more integrated, adaptive, and intelligent robotic systems. The incorporation of multimodal data and end-to-end learning approaches has significantly enhanced the capabilities of these systems, enabling them to perform a wide range of tasks with higher accuracy and contextual understanding. Future research is likely to build on these foundations, exploring new ways to further integrate and optimize these models for even more sophisticated applications.

# 3 METHOD

## 3.1 PROBLEM STATEMENT

**Challenges to be overcome.** While current models have achieved embodied intelligence at high- and mid-level control layers, few models address the low-level control layer. Human needs at the high- and mid-levels are explicit, easily articulated by users. However, low-level control needs are implicit and difficult to express as data(Figure 2) , resulting in a lack of training data that hinders model performance validation. Consequently, low-level control models cannot be trained directly based on feedback.

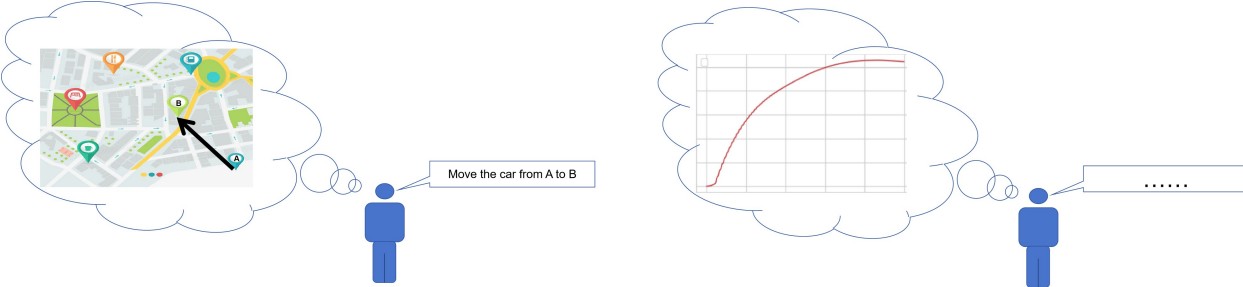

Figure 2: Humans can clearly articulate their requirements for high-level control. However, they are unable to precisely express their needs for low-level control.

To address this, we propose leveraging large language models (LLMs) to simulate human needs. Since LLM-generated low-level needs are explicitly defined by humans, this approach overcomes the data scarcity issue. Furthermore, by inputting both the control method mechanisms and their outcomes into the LLM, the model can understand the effects of control strategies and enable low-level human-robot interaction, fostering true empathy in robots, namely,enhancing robots' empathy.

**Definition.** In human-robot interaction, dynamic environments require that robots adjust their control behavior based on human feedback. The definition of information encompasses all perceptible data to humans, including but not limited to temperature, acceleration, force, and other sensory inputs.The information $I$ acquired by humans from the environment is influenced by robot controller parameters $P$, time $T$, and environmental conditions $E$, described as:$I = f(P, T, E)$ To simplify the problem, we consider a specific time interval $T_0 = [t_0, t_1]$ and a particular environment $E_0$, resulting in the information $I_0$ being expressed solely as a function of the controller parameters:$I_0 = f(P)$ This formulation allows us to focus on the impact of robot control decisions on human perception within a defined context.

Human perception of environmental information varies across individuals, resulting in different sensations $S$. This relationship can be modeled as:$S = f(I)$ Moreover, perception depends on individual differences, which introduces a dependency on the person $H$, such that:$S = f(I, H)$ For a specific individual $h$, this can be simplified to:$S_h = f(I)$ These variations in perception necessitate the adaptation of control strategies to individual preferences.

Conventional control methods generally tune the parameters $P$ based on aggregate user perception, with limited subsequent adaptation, resulting in suboptimal performance for personalized human-robot interaction.

$$P = \arg \max_P \left( \sum_H S_h(I) \right) \tag{1}$$

The objective of this work is to personalize the adjustment of controller parameters $P$ based on individual preferences, optimizing for each person's sensation $S_h(I)$. Formally, this is represented as:

$$P_h = \arg \max_P S_h(I) \tag{2}$$

This approach enhances the robot's ability to adapt to real-time feedback, providing more intuitive and effective human-robot interactions. However, it also introduces challenges in efficiently capturing and processing individual feedback.

## 3.2 ENHANCING ROBOTS' EMPATHY

This study aims to enhance robots' empathy by enabling the modulation of the robot's actions based on human instructions. This approach allows the robot to adjust its actions dynamically, reflecting an empathetic understanding of human preferences.

The algorithm is designed to optimize robot control parameters by utilizing feedback from human operators. Initially, the environmental conditions and parameter settings are defined. The process involves pre-training a large language model to recognize the relationships between control parameters and the resulting performance. Subsequent steps involve:

Figure 3 and Algorithm 1 illustrate the mechanism for enhancing robot empathy, showcasing how robots can dynamically adapt their actions based on human feedback.

## 3.3 GENERATING SIMULATED DATA

To generate the required training data, we simulate information $I_0$ concerning initial parameters $P_0$ within a simulation environment $E_0$ over a specified time period $T_0$. Assuming an individual's preference is $P_{\max}$, information $I_{\max}$ is simulated under the parameters $P_{\max}$. The large model employs prompt engineering to compare $I_0$ with $I_{\max}$, generating corresponding sensations $S$ and descriptions $D$. Initially, the model is pre-trained and fine-tuned as necessary. An initial parameter $P_0$ is preset, and within the simulation environment $E_0$ during the time period $T_0$, the robot executes actions to gather information $I_0$. Concurrently, actions corresponding to $P_{\max}$ are executed to obtain information $I_{\max}$. The model $M_h$ then generates sensations $S_{0h}$ based on the information $I_0$ and $I_{\max}$, and subsequently describes these sensations to obtain description $D_0$. The model $M$ receives the sensations described in $D_0$ and adjusts the model parameters to derive new parameters $P_1$. This iterative process is repeated until model $M_h$ is satisfied with the parameters $P_h$.

---

**Algorithm 1** Optimizing Robot Controller Parameters Based on Individual Human Sensations

---

1: **Input:** Initial environment $E_0$, time interval $T_0$, initial parameter $P_0$
2: **Output:** Optimized parameter $P_h$
3: Pre-train and fine-tune a large language model $M$
4: **for** each control method **do**
5:     Prompt model $M$ to understand the relationship between control parameters and feedback
6: **end for**
7: Set initial parameter $P_0$
8: **repeat**
9:     Execute robot actions in $E_0$ during $T_0$ to collect data resulting in $I_0 = f(P_0)$
10:     Human $h$ experiences $I_0$, producing sensation $S_{0h} = f(I_0)$
11:     Human $h$ describes their sensation to obtain $D_0$
12:     Model $M$ receives $D_0$ and modifies the model parameters to derive new parameter $P_1$
13:     Update $P_0$ to $P_1$
14: **until** Human $h$ is satisfied with the parameter $P_h$

---

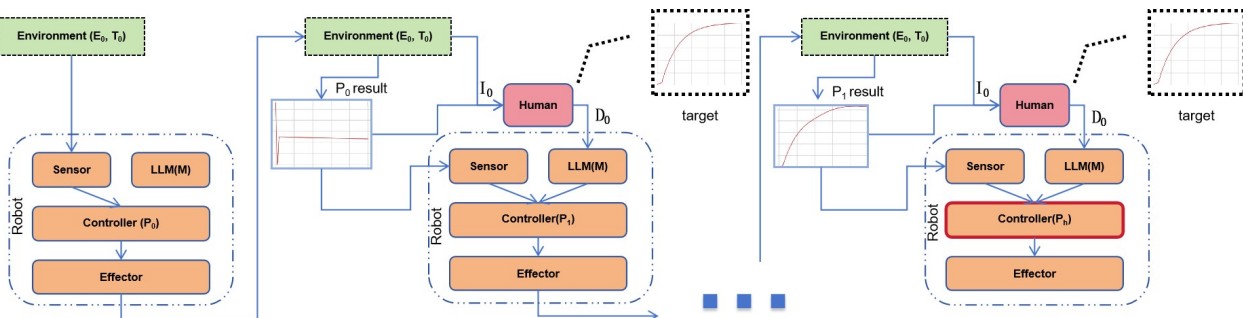

Figure 3: Mechanism for enhancing robots' empathy

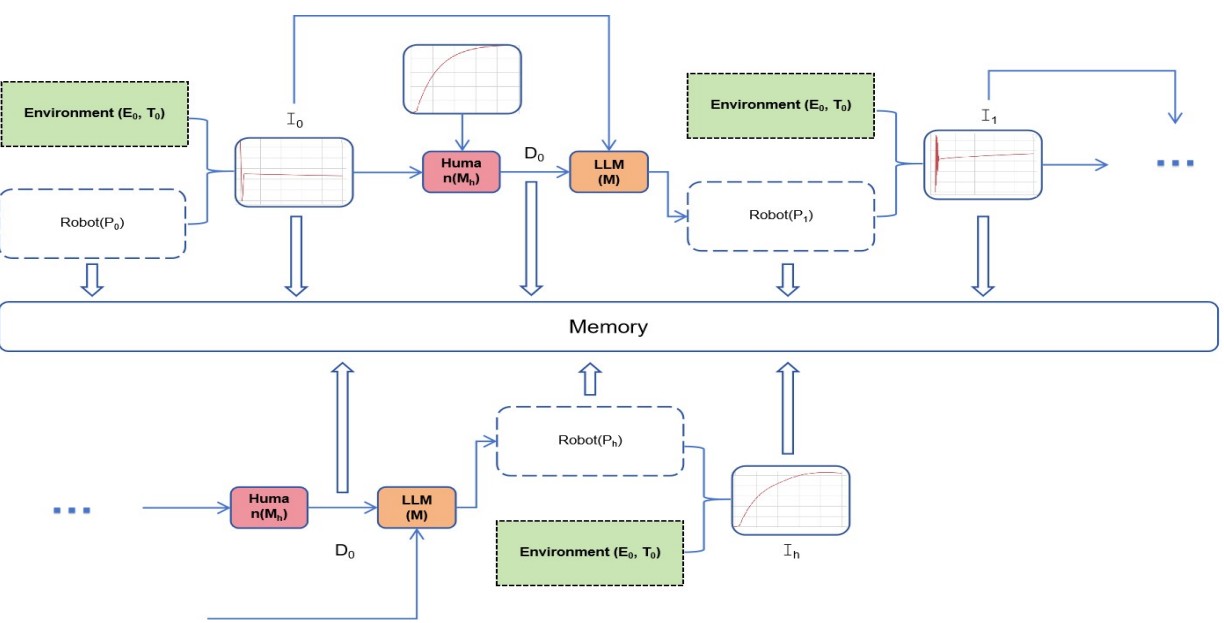

Figure 4: Mechanism for generating simulated data

This method generates a substantial amount of simulated data for model training, enhancing the model's accuracy and performance. This approach leverages the capabilities of large language models to handle time series data, effectively addressing the scarcity of human feedback data.

---

**Algorithm 2** Generating Simulated Data for Model Training

---

1: **Input:** Initial environment $E_0$, time interval $T_0$, initial parameter $P_0$
2: **Output:** History of parameters and model outputs
3: Pre-train and fine-tune the large language model $M$ to adapt to control feedback, applying prompt engineering techniques.
4: Initialize history logs to record information $I$, parameters $P$, and outputs from models $M_h$ and $M$.
5: Set initial parameter $P_0$.
6: **while** a termination condition is not satisfied **do**
7:     In environment $E_0$ and during time interval $T_0$, command the robot to perform actions that yield information $I_0 = f(P_0)$.
8:     Simultaneously, execute actions under $P_{\max}$ settings to generate information $I_{\max}$.
9:     Based on $I_0$ and $I_{\max}$, model $M_h$ computes sensation $S_{0h} = f(I_0, I_{\max})$.
10:    $M_h$ describes its computed sensation to produce description $D_0$.
11:    Model $M$ processes $D_0$ to adjust the parameters, deriving a new set $P_1$.
12:    Log $I_0$, $P_0$, $D_0$, and $P_1$ into the history.
13:    Update $P_0$ to $P_1$.
14: **end while**
15: **Return:** History documenting the progression of parameters and their corresponding outputs.

---

Algorithm 2 and Figure 4 demonstrate the mechanism for generating data.

### 3.4 HOW TO DEFINE EMPATHY

The proposed concept of empathy in this paper includes two aspects. First, the robot should better understand human intentions and perform actions that align with human needs within its capabilities, without being restricted by any pre-set parameters. Second, the agent should think from the human perspective, explicitly displaying preferences for action styles, which are difficult to quantify, through curves or other forms. This enables the batch generation of training data.

First, current training for robot actions emphasizes whether the robot has completed a target task, such as picking up an object or reaching a destination. However, the process of how the robot accomplishes the task has been largely overlooked. The robot's control methods are typically pre-defined, and the actions taken by the robot while completing the task are constrained by these control parameters. The fixed parameters of the control methods limit the robot's actions to a subset of its action space. As a result, during task execution, the robot cannot dynamically adjust its control methods in real-time based on user preferences. This paper proposes a method that allows robots to dynamically adjust their control parameters in real-time, which differs from previous methods that adjust the reward function to alter the robot's action style. Our method can be applied directly and in real-time to the robot, allowing users to adjust the robot's action style dynamically during interactions without specialized knowledge. This has significant potential in fields such as healthcare and caregiving.

Second, human preferences for action styles are difficult to quantify. In the field of embodied intelligence, current methods for evaluating models typically assess task completion rates. However, action style preferences cannot be evaluated using this metric. One approach is to use binary satisfaction criteria, where human evaluators experience and rate their satisfaction. However, this requires separate data collection for each task and action, which is impractical in practice. To address this challenge, we propose using a large language model as the agent to generate large amounts of data for multiple tasks and actions. In this framework, the language model simulates human preferences for action styles by pre-setting optimal parameters. This approach allows for the generation of training or evaluation data in various scenarios involving different control methods, tasks, and actions.

In summary, the robot's action control problem can be divided into two parts: the search for control method parameters, and the execution of actions once the control method is determined. This paper proposes a framework for human-robot interaction in the adjustment of control method parameters.

All robot actions are executed by power sources, such as motors, with the combined output from the motors at each joint constituting the robot's motion. Therefore, we define the space of motor outputs as $A_0$, which encompasses all possible actions of the robot. However, in practical applications, the control methods of the robot are predefined, and thus, the set of robot actions becomes a subset $A_p$ of $A_0$ under the constraints imposed by the control parameters $p$ (Figure 5a). The desired action styles for different users may lie within different subsets $A_p$. Through dynamic

parameter tuning, the constraints can be overcome, allowing the optimal solution to be found across the entire feasible space. As shown in Figure 5b , The current work primarily focuses on Stage II, while this paper emphasizes Stage I.

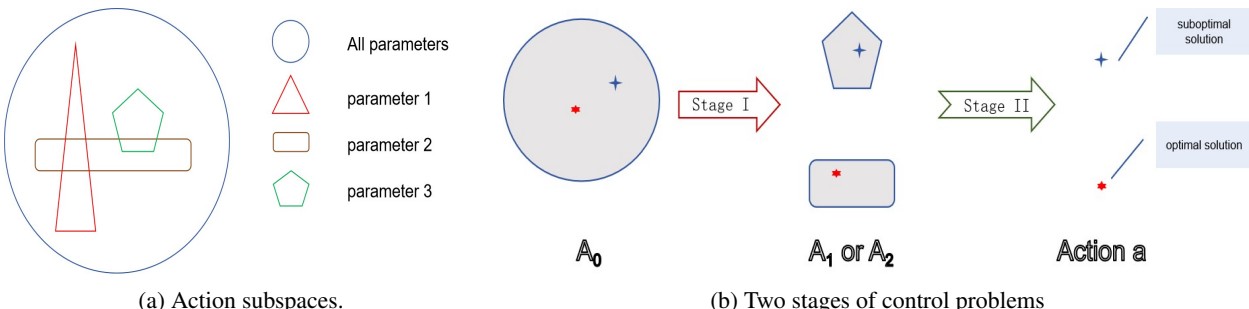

(a) Action subspaces.       (b) Two stages of control problems

Figure 5: The control problem is divided into two stages: parameter tuning of the control method and action selection.

## 4  EXPERIMENTS

This paper conducts experiments on two control methods: PID and NMPC, both of which are commonly used algorithms in the field of robotic control. Two types of problems were designed for these methods: a simple position control problem for PID, and a trajectory tracking problem for NMPC. In both cases, parameters were adjusted multiple times during problem setup to ensure the generality and robustness of the results.

### 4.1  SIMULATION ENVIRONMENT

**Proportional-integral-derivative (PID).** This work addresses the position tracking of a one-dimensional mobile system, aiming to bring the system to a desired target position and ensure it stops there. The dynamics of the system are influenced by control force $f(t)$, external force $F(t)$, friction characterized by the coefficient $\mu$, and mass $m$. The system's acceleration $a(t)$ is governed by:

$$a(t) = \frac{1}{m}\left(f(t) + F(t) - \text{sign}(v(t)) \cdot \mu \cdot m \cdot g\right) \tag{3}$$

The variables used in the system dynamics are defined as follows: $f(t)$ is the control force generated by the PID controller, $F(t)$ is the external force or disturbance acting on the system, $\mu$ is the coefficient of friction, $m$ is the mass of the system, $g = 9.8\,\text{m/s}^2$ is the gravitational constant, and $\text{sign}(v(t))$ is the sign function that determines the direction of velocity $v(t)$, and consequently, the direction of the frictional force.

The control objective is to minimize the position error $e(t) = x_{\text{target}}(t) - x(t)$, where $x(t)$ is the current position and $x_{\text{target}}(t)$ is the target position. The PID controller dynamically adjusts the control force $f(t)$ to achieve this. The system's velocity $v(t)$ and position $x(t)$ are updated at each time step $\Delta t$ using the following discrete-time equations:

$$\begin{aligned} v(t + \Delta t) &= v(t) + a(t) \cdot \Delta t \\ x(t + \Delta t) &= x(t) + v(t) \cdot \Delta t \end{aligned} \tag{4}$$

The control force $f(t)$ is subject to a maximum allowable force $f_{\text{max}}$, ensuring safe and stable operation. The force is constrained as follows:

$$f(t) = \begin{cases} f_{\text{max}} \cdot \frac{f(t)}{|f(t)|}, & \text{if } |f(t)| \geq f_{\text{max}} \\ f(t), & \text{otherwise} \end{cases} \tag{5}$$

Friction, modeled as $F_{\text{friction}} = \mu \cdot m \cdot g$, opposes the system's velocity, helping to decelerate the system and stop it at the desired position. The PID controller ensures that the system decelerates smoothly as it approaches the target, bringing the velocity $v(t)$ to zero at the target position.

**Nonlinear model predictive control (NMPC).** The dynamics of a car-like mobile robot (CMR)(Figure 6) are utilized to validate the performance of the intelligent controller within the framework of nonlinear model predictive control

(NMPC).Ding et al. (2021) The system is represented by the following nonlinear kinematic equations: The kinematic model of the CMR is described by the following set of equations:

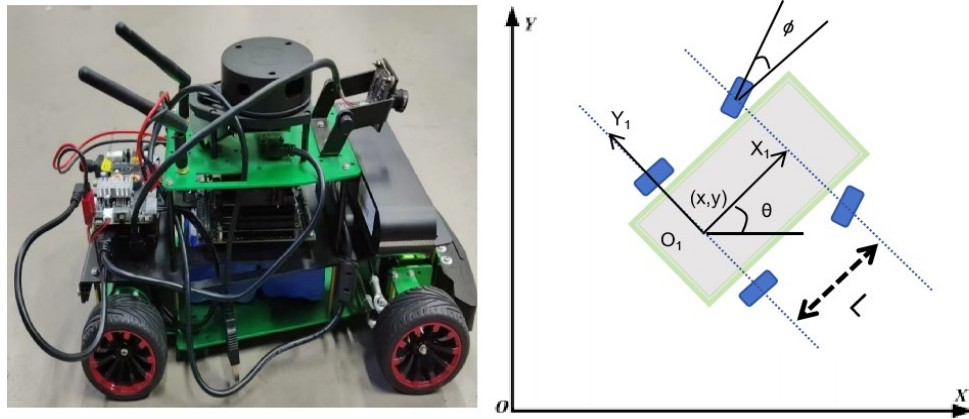

Figure 6: Car-like robot in our lab and its simplified physical model.

$$\dot{x} = v \cos \theta, \dot{y} = v \sin \theta, \dot{\theta} = \frac{v}{L} \tan \phi, \tag{6}$$

where $x$ and $y$ are the robot's position coordinates, $\theta$ is the heading angle, $v$ is the linear velocity, $L$ is the wheelbase, and $\phi$ is the steering angle of the front wheels.

To design control strategies, the tracking error between the CMR and a reference trajectory is governed by the following error dynamics:

$$\dot{e}_x = v \cos \theta - v_r \cos \theta_r, \dot{e}_y = v \sin \theta - v_r \sin \theta_r, \dot{e}_\theta = \frac{v}{L} \tan \phi - \frac{v_r}{L} \tan \phi_r, \tag{7}$$

where $e_x = x - x_r$, $e_y = y - y_r$, and $e_\theta = \theta - \theta_r$ represent the position and angle errors, while $v_r$ and $\phi_r$ denote the reference velocity and steering angle, respectively.

## 4.2 RESULTS

In both scenarios, we experimented with various parameters to demonstrate the robots' capability to adapt control strategies in response to human requests. To visually evaluate the performance of the proposed method, we employ the Fréchet distance to quantify the similarity between control curves.

The *Fréchet distance* is a metric designed to compare two curves.

Consider two continuous curves $P : [0, 1] \to \mathbb{R}^d$ and $Q : [0, 1] \to \mathbb{R}^d$, parameterized over the interval $[0, 1]$. The Fréchet distance between $P$ and $Q$ is defined as:

$$d_F(P, Q) = \inf_{\alpha, \beta} \max_{t \in [0,1]} \| P(\alpha(t)) - Q(\beta(t)) \|, \tag{8}$$

where $\alpha$ and $\beta$ are continuous, non-decreasing reparameterizations of the interval $[0, 1]$.

**Proportional-integral-derivative (PID).** When evaluating the intelligent controller on the PID method, we varied environmental parameters such as the cart's mass, friction coefficient, initial position, and initial velocity ( Table 1). The framework was employed to generate human preferences, enabling the controller to intelligently adjust the three PID parameters in response to these preferences. The results are shown in Figure 7b, where it can be observed that the adjusted control strategy closely matches the target control strategy across multiple trials. Additionally, the Fréchet distance (Figure 7a) between the error curve of the control method and the target curve decreases consistently, providing further validation of the method's effectiveness.

**Nonlinear model predictive control (NMPC).** When evaluating the intelligent controller on the NMPC method, four distinct cases(Table 2) were designed to validate the general applicability of the approach.The first parameter represents the importance ratio of the error in the Y direction to the error in the X direction, while the second parameter

Table 1: Description of the five experiments in the simulation.

| Experiment ID | Position (pos) | Force (F) | Friction coeff. (mu) | Mass (kg) | Initial velocity (v) |
|---|---|---|---|---|---|
| 1 | 1 | 2 | 0.01 | 20 | 0.1 |
| 2 | -1 | 5 | 0.03 | 2 | 0.1 |
| 3 | 0 | 5 | 0.03 | 2 | 1 |
| 4 | -2 | -5 | 0.015 | 5 | 1 |
| 5 | 1 | 0 | 0.01 | 10 | 0 |

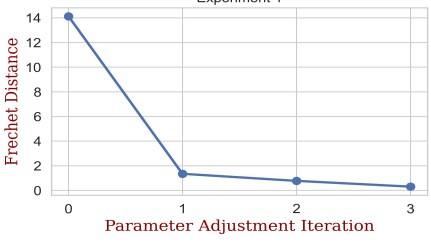

(a) Fréchet distance.

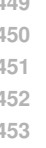
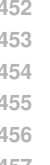
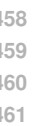
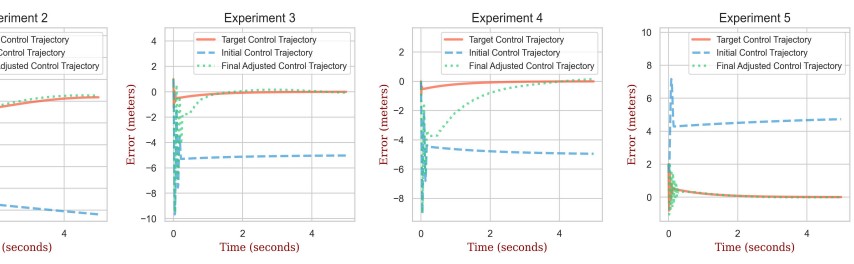

(b) PID errors over the time.

Figure 7: PID Results: Fréchet distance(experiment 1) and errors over time.

represents the importance of the angular error.The framework generated human preferences to allow the controller to intelligently adjust the NMPC parameters in response to these preferences. The results, as shown in Figure 8a, demonstrate that the adjusted control strategy closely aligns with the target control strategy after multiple trials. Furthermore, the Fréchet distance (Figure 8b) between the error curve of the control method and the target curve consistently decreases, offering additional evidence for the effectiveness of the proposed method.

Table 2: Target and initial parameters for NMPC validation across four cases.

| Case ID | Target Parameters | Initial Parameters |
|---|---|---|
| 1 | [0.05, 0.03] | [10, 0.01] |
| 2 | [0.05, 0.03] | [3, 0.07] |
| 3 | [2, 0.3] | [7, 0.4] |
| 4 | [3, 0.1] | [7, 0.4] |

**Conclusion.**The experimental results demonstrate that the proposed framework is capable of training intelligent models that can fine-tune various robotic control methods across diverse environments to meet human requirements. Additionally, the framework can emulate human behavior to generate language data(e.g., Appendix A.1), further enhancing its applicability in human-robot interaction tasks.

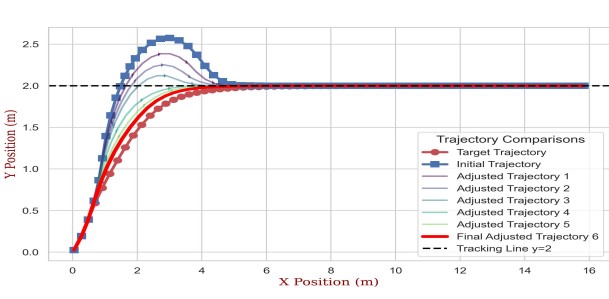

(a) NMPC trajectory tracking: case 1 results.

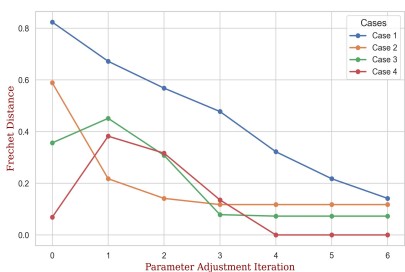

(b) Fréchet distance over parameter adjustments for all cases.

Figure 8: NMPC Results: Trajectory tracking(case 1) and Fréchet distance for all cases.

# 5 CONCLUSION AND DISCUSSION

This work presents an end-to-end embodied intelligence framework that integrates large language models (LLMs) with control algorithms, improving human-robot interaction by translating human inputs directly into robotic actions. Initially demonstrated using a PID control algorithm, the proposed model simplifies the interaction between humans and robots, laying the groundwork for more intuitive control systems.

Our approach extends beyond PID control, as we successfully implemented intelligent parameter adjustment within a Nonlinear Model Predictive Control (NMPC) framework. This allows the control system to dynamically adapt NMPC parameters based on human instructions, aligning control behavior with user intentions. This development demonstrates that LLMs can facilitate more sophisticated control strategies, enhancing system flexibility and responsiveness.

A key contribution of this research is bridging the gap between high-level decision-making and low-level control. By enabling LLMs to influence both planning and real-time action control, the proposed system allows robots to fine-tune their actions in response to dynamic human inputs. This advancement marks a significant step forward in the field of embodied intelligence, enabling robots to directly adapt their control dynamics based on user preferences.

Methodologically, we leveraged LLMs with prompt engineering to process time-series data and generate synthetic training data, addressing the common challenge of acquiring large-scale human-annotated data. This scalable framework enables more efficient batch generation of synthetic data for training embodied intelligence systems, demonstrating the feasibility of using LLMs in control tasks.

However, while the effectiveness of our approach has been validated in simplified dynamic systems, several limitations must be addressed for broader applicability. First, integrating LLMs with more complex control methods, such as sliding mode control (SMC) Utkin (1977) or Proportional-Integral-Derivative Neural Networks (PIDNN) Yu et al. (2016), requires further exploration. These advanced algorithms may necessitate more sophisticated prompt designs and rigorous testing in diverse operational contexts to ensure scalability and robustness. Additionally, the computational demands of using LLMs in low-level control tasks must be carefully evaluated, particularly in time-sensitive applications where latency could impact performance.

Future research should focus on optimizing the computational efficiency of the framework and expanding its applicability to more complex, real-world environments. Specifically, further studies are needed to explore the scalability of LLM-based control systems in handling dynamic, nonlinear environments and the integration of advanced control algorithms. Addressing these challenges will be crucial for realizing the full potential of LLMs in enhancing robotic control and interaction capabilities.

Overall, our findings suggest that LLMs, traditionally applied to language understanding and high-level decision-making, can play a critical role in directly controlling robotic systems. This research opens new avenues for embodied intelligence, demonstrating that language-based models can be integrated with advanced control algorithms to create more adaptive, responsive AI-driven systems. Further investigation is needed to validate this approach in more complex and dynamic environments, paving the way for the next generation of intelligent robotic systems.

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

# A APPENDIX

## A.1 A PART OF HISTROY (PID EXAMPLES)

Below is part of the data generated during the PID experiment, including the original control parameters, human instructions imitated by the LLM, the robot's thought process, and the new control parameters.

controller:[10, 0.1, 1]

errors:[-1.0, -1.0, -0.9991295526330, -0.8934778552241, -0.8209091474065, -0.732043664220, -0.6711492358760, -0.6024086502903, -0.545791766502, -0.497873424167, -0.4463078279172, -0.4072604387097, -0.3635477452604,

-0.3339659798670, -0.29939440195804, -0.2725427572730, -0.24440932437141, -0.22322850636001, -0.20081121961140,

-0.18135977989241, -0.16232019893271, -0.143314073981, -0.12452858309592, -0.10647938553509, -0.08907150003000,

-0.07253160223958, -0.05667596460464, -0.0415751874602, -0.027903696622094, -0.015378272403062, -0.003909658878785,

0.00658183334035,

0.015900407475726, 0.024271477980404, 0.03167099820390, 0.037791089474880, 0.04332661547359, 0.04777156101173,

0.05176149725117, 0.054641784895456, 0.056660755954281, 0.057936097068903, 0.05867785059124, 0.059470207751414,

0.05882543416812, 0.05759230644816, 0.05610944454285, 0.05412389005588, 0.051710528072810, 0.04911821638070] human:Try to smooth out the changes, reduce the fluctuations, and speed up getting to the target. Focus on making your actions more stable and precise. adviser:The current error values demonstrate a significantly slower rate of convergence and show more pronounced fluctuations compared to the expected error trajectory. There are also noticeable differences in the intensity of the changes and the speed of convergence between the current error and the expected values.

In terms of frequency of fluctuations, the current error values show more frequent and larger fluctuations, especially in the early stages, compared to the expected trajectory.

In terms of intensity of changes, the current error values exhibit larger and more abrupt changes, especially in the early stages, compared to the expected values, which show more gradual and smooth changes.

In terms of speed of convergence, the current error values converge more slowly and take longer to approach the expected target values, compared to the expected trajectory, which reaches the target values more quickly.

To address these differences, the controller should be adjusted to minimize fluctuations and reduce the intensity of changes, while speeding up the convergence towards the expected target values. This can be achieved by modifying

the control parameters to make the adjustments more gradual and precise, and by emphasizing stability and smoothness in the control strategy.

controller:[1, 0.01, 0.5]

