# OpenReview forum: "Intelligent Control in Embodied Robotics: Enhancing Human-Robot Interaction through Adaptive Control Techniques"
_ICLR.cc/2025/Conference — Submitted to ICLR 2025_

### Official Review · Reviewer_FC5y · 2024-10-31

**Soundness:** 2
**Presentation:** 1
**Contribution:** 2
**Rating:** 3
**Confidence:** 4

**Summary:**

This paper addresses the challenge of enabling robots to dynamically adapt their low-level control parameters in response to human intentions—a limitation in current embodied intelligence models. The authors propose a framework that uses large language models (LLMs) to directly optimize controller parameters while keeping human feedback in the loop as text prompts to the LLM.

The authors demonstrate the framework on two classic controllers—PID and Non-Linear MPC—on a robot car, showing that the proposed framework is capable of outputting and optimizing low-level controls that match human commands/preferences.

**Strengths:**

**Originality:**

- The paper focuses on translating high-level commands to low-level control using LLMs in the domain of human-robot interaction, which is a novel topic in recent years.

**Quality:**

- The paper conducts full experiments on two types of controllers on a physical robot, with quantitative data analysis demonstrating that the proposed framework is capable of optimizing robot controllers given human feedback.

**Clarity:**

- The use of diagrams and figures helps explain the proposed framework and the overall challenge of translating high-level commands to low-level control, highlighting why the problem is important.

**Significance:**

- The paper tackles the significant issue of adaptability in low-level robot control, which is crucial for enhancing human-robot interaction.

**Weaknesses:**

**Originality:**

- Although the topic is novel, there are several established approaches that bridge the gap between high-level commands and low-level control through the use of large language models (LLMs) on low-level controllers, making the authors' work not entirely new.
    - **Yu, Wenhao, et al. "Language to rewards for robotic skill synthesis." *arXiv preprint* arXiv:2306.08647 (2023):**
        - This paper introduces a framework in which LLMs translate natural language instructions into reward functions. These functions are then optimized by a motion controller (e.g., RL or MPC) to generate low-level control actions, demonstrating complex skills like making a robotic dog perform a handstand or a moonwalk.
    - **Ma, Yecheng Jason, et al. "Eureka: Human-level reward design via coding large language models." *arXiv preprint* arXiv:2310.12931 (2023):**
        - This paper introduces a framework that translates natural language commands into reward functions, with a focus on robot skill acquisition via reinforcement learning (RL) rather than on optimizing PID or MPC for human-robot interaction.
    - While the current paper’s approach is unique in that it directly prompts LLMs to output control parameters in textual form (as opposed to using an intermediate reward function), it still overlaps with previous work in that both approaches translate high/mid-level commands to low-level control through LLMs.
        - The authors could mention this existing work and clarify their approach’s uniqueness by emphasizing the absence of an intermediate reward representation.
        - They might also find inspiration in similar works that employ LLMs as optimizers, such as:
            - **Yang, Chengrun, et al. “Large Language Models as Optimizers.” *arXiv preprint* arXiv:2309.03409 (2024):**
                - This work uses LLMs iteratively to generate solutions for optimization tasks, like linear regression and the traveling salesman problem, by updating prompts to improve solutions, which could offer useful insights for the authors.

**Quality:**

- The integration of LLMs with control algorithms is insufficiently detailed. It is unclear how the LLM processes human feedback and translates it into control parameter adjustments. Specifically:
    - From the conversation example in the appendix, it can be inferred that the LLM outputs control parameters directly in text form, with human preferences/instructions added to the prompt. However, this is not clearly explained in the main article.
    - To improve clarity, the authors could create or update diagrams that illustrate the human-interaction schema, clarify data modalities at each step, and provide examples of human commands.
- The human commands used in experimental validation are limited to simple dynamics (e.g., “speed up,” “reduce fluctuation”).
    - This is inferred from the example in the appendix, as the authors did not describe the types of human commands/preferences used in the experiment.
        - They could provide more examples of human-robot interaction or include more detailed system architecture diagrams.
    - The focus on simple dynamics like “speed up” is problematic, as these are common optimization objectives. It remains unclear whether the optimization is genuinely guided by human preferences or if it is merely performing basic optimization tasks.
        - This argument would be strengthened if the authors demonstrated uncommon objectives like “spin around” or “move in a zigzag motion.”
- The paper does not quantitatively compare the proposed method against existing approaches, making it difficult to evaluate the contributions' significance.
- The paper mentions robot empathy but does not define what empathy means in this context or provide quantitative measurements and evaluation.
    - Based on the work described, it appears the authors interpret empathy as the robot's ability to adjust control output based on human feedback. However, empathy as a concept is broader (encompassing emotion, theory of mind, etc.).
    - A clear, limited definition of empathy would strengthen the paper.

**Clarity:**

- The paper discusses broad topics such as embodied intelligence and human-robot interactions, which may be too general and not directly relevant to the work presented.
    - Based on the actual experiment, the authors could consider limiting the scope to focus on translating mid-level commands to controller output and optimizing with LLMs based on human preferences. They could also discuss background and related work specifically in this area in the introduction section, rather than broadly covering topics like embodied intelligence, human-robot interaction, and empathy.
- The authors should clarify which LLM model they are using, and what techniques (e.g., prompt engineering, fine-tuning) are applied. Adding these details, with citations, would improve clarity.
- The paper lacks sufficient details and examples of the types of human commands/preferences being incorporated.
- Several parts of the paper are unclear, with grammatical errors and missing space characters. Examples include:
    - "thus Enhancing Robots’ Empathy(ERE)."
    - "layersFigure 1"
    - "physical worldPfeifer & Iida (2004)"
- Section titles like “Difficulties” and “Problem” could be more specific.
- Including a "Conclusion" section within the "Related Work" section is unusual and could be reconsidered.

**Significance:**

- Although using LLMs to adapt low-level control parameters is promising, the experiments in the paper are relatively simplistic and may not convincingly demonstrate the practical significance of the proposed method. The tasks used for validation—such as adjusting simple dynamics in a robotic car—are basic and do not fully showcase the potential benefits of the approach in more complex or real-world scenarios. This limits the ability to assess how the method would perform in more complex and uncertain settings, which are common in practical human-robot interactions.
- Without quantitative comparisons with existing methods, it is challenging to evaluate the significance of the proposed approach.
- The paper does not sufficiently discuss how the proposed framework could generalize to other types of robots, control methods, or tasks beyond the one tested. Without demonstrating broader applicability, the significance of the work may be limited to niche applications.
- Since the paper emphasizes improving human-robot interaction and robot empathy, the absence of user studies or evaluations involving human participants is a significant gap.

**Questions:**

- Could the authors provide more detailed explanations or examples of how the LLM processes human feedback and adjusts control parameters? Specifically, how does the LLM interface with the control algorithms? What types of human feedback and commands are being used besides the one provided in the appendix? Can the authors provide an overall description?
- Have the authors considered conducting experiments with more complex robots and on more challenging tasks?
- Can the authors perform human evaluation studies to see how well the proposed framework addresses human feedback/intention and how well it improves robot empathy?
- How does the proposed method compare quantitatively with existing approaches that address adaptability in robot control and translate high-level commands to low-level control? Including such comparisons would strengthen the evaluation.
- What are the computational requirements of integrating LLMs into low-level control tasks? How does the method ensure real-time performance?

---

> ### Author Response · Authors · 2024-12-02
>
> Thank you very much for your suggestions. Your feedback was detailed and provided us with valuable insights. We have included the articles you recently mentioned in the paper, revised some of the section titles, and added a new section to describe what empathy is.
> Q1: In brief, the LLM receives natural language instructions provided by humans, then generates new parameters based on those instructions, which are subsequently applied to the controller. In the simulations presented in this paper, the LLM directly invokes pre-defined functions to adjust the control parameters. Human feedback is solely conveyed through natural language, which simply describes the user's requirements.
> Q2: We are currently developing a simulation environment that will allow users to define various control methods and robot models for training purposes. However, due to time constraints, results have not yet been provided.
> Q3: We employ a distance metric to represent the discrepancy between the robot's control performance and the desired target control performance, using it to evaluate the effectiveness of the framework. Our findings indicate that the robot can achieve the target control performance after just a few adjustments.
> Q4: The primary difference between the current method and existing approaches is that it allows users to adjust the robot's control style according to their needs without requiring specialized knowledge. Furthermore, we provide a quantitative method to assess whether the robot's control style aligns with human requirements.
> Q5: The method can be applied to small-scale models and is compatible with consumer-grade graphics cards, such as the 4090. Each parameter adjustment can be completed within a few minutes.

---

### Official Review · Reviewer_CaUV · 2024-11-04

**Soundness:** 3
**Presentation:** 3
**Contribution:** 3
**Rating:** 5
**Confidence:** 4

**Summary:**

This paper introduces a novel framework to realize  human-in-the-loop improvement of robot behavior by allowing robots to dynamically adjust their control strategies based on real-time human feedback.
The framework utilize large language models (LLMs) and adaptive control methods.
 By using LLMs for simulated data generation, it achieves a new type of personalized human-robot interactions.
The method is combined with PID and NMPC control.
The performance was tested using a simple robot system,

**Strengths:**

- The paper innovatively uses LLMs to adapt control parameters for low-level robot control, whereas most LLM applications in robotics focus on high-level planning. This novel approach to using LLMs for low-level controller adjustment represents a significant contribution.
- The proposed framework enables real-time personalization, offering a fresh and promising approach to human-robot interaction.

**Weaknesses:**

- he framework has been tested primarily in simulated and simplified environments, demonstrating only preliminary validation of the proposed concept. The paper lacks testing in realistic robotic scenarios, which limits the strength of evidence supporting its practical applicability.

**Questions:**

<Major Comments>
1. How can this framework be extended and validated for more complex and realistic robotic scenarios?
2. The advantages of using LLMs for parameter tuning over active exploration methods (like Bayesian optimization) need to be better demonstrated.



<Minor Comments>
1. Several parentheses are missing throughout the text, for example:
   - "worldPfeifer & Iida, 2004 (Line 32)
   - "layersFigure 1 (Line 53)

2. Figure 1 depicts low-level control based on position control. For real-world applications, dynamic aspects such as force-based and velocity-based control are crucial. Given the citations of Pfeifer et al.'s work, the paper should address dynamics and morphological computation.

3. Figure 5 appears distorted.

4. The definition of I (information) lacks clarity.

5. Figures 6 and 7 have illegible legends.

---

> ### Author Response · Authors · 2024-12-02
>
> Thank you for your comments. We have corrected the formatting errors in the article and added a more detailed explanation of empathy.
>
> Q1:
> Because our framework can be applied to any large language model, the effectiveness of this method will continue to improve with the advancement of models. We are currently working on building a simulation environment based on Habitat, allowing different researchers to train various control methods on it, and adapt to various customized scenarios. However, due to time constraints, this has not yet been completed.
> Q2:
> We have added a section, "How to Define Empathy," to explain the timeliness of the method. Compared to traditional methods, our approach allows users to adjust parameters in real-time without requiring specialized knowledge.

---

### Official Review · Reviewer_pfeG · 2024-11-04

**Soundness:** 2
**Presentation:** 3
**Contribution:** 2
**Rating:** 3
**Confidence:** 4

**Summary:**

The paper proposes using a prompting mechanism with a large language model (LLM) to fine-tune parameters of two control algorithms to align with human preferences. It attempts to introduce "empathy" as a guiding concept in algorithmic adjustments.

**Strengths:**

1. Uses LLMs, a current trend, which may attract interest.
2. Provides a rudimentary exploration of aligning algorithm outputs with user preferences.

**Weaknesses:**

I think works lack novelty. Using a large language model (LLM) to adjust control parameters based on human preferences is a repurposing of existing techniques rather than a novel concept. Human-in-the-loop control systems and preference-based tuning have been well-explored, making this approach more about applying known methods than advancing new knowledge. It doesn't break new ground conceptually, which is crucial for meaningful research contributions.

The paper then introduces empathy as a goal without a clear definition or a rigorous way to measure it in a control context. Empathy is not inherently quantifiable in control algorithms, and this lack of clarity makes the problem ill-suited for rigorous scientific investigation. While this might offer an interesting discussion for a student project, research demands concrete metrics and definitions, which this project does not provide. The use of this term seems more like a buzzword than a meaningful contribution.

Without a specific, compelling application or demonstrable impact, this type of control tuning has limited relevance. The problem is more theoretical than practical, with no strong justification for why fine-tuning to human preferences adds substantial value or solves a pressing issue. This scope is acceptable for student exploration but lacks the depth and relevance expected in publishable research.

I also find some vagueness in methodology: How human preferences are generated and quantified is unclear. The authors do not provide a rigorous methodology for capturing, validating, or generalizing these preferences. This makes the approach seem arbitrary and undermines reproducibility.

**Questions:**

How were human preferences quantified, and what criteria were used to validate these preferences?
What concrete advantages does this approach offer over traditional or established methods in adaptive control?
Can the authors clarify the role of "empathy" in this study? How does it translate to actionable parameters in control algorithms?

---

> ### Author Response · Authors · 2024-12-02
>
> Your analysis is very thorough, and based on your suggestion, we have added a more detailed explanation of empathy.
> Q1:
> This is a key question. We set the control methods’ parameter human prefer and give the control result to the LLM.  We use a distance between the current control curve and the preferred control curve to validate the adjustment result.
> Q2:
> This approach is fast and enable those who do not understand how to adjust the control parameters to get a new control style easily.
> Q3:
> We add a new part at the paper to explain ‘empathy’ . It actually contains two part: robots can understand humans demands and adjust control style in time; and agents given target parameters can act as human to give opinion to current control methods.

---

### Official Review · Reviewer_Y2be · 2024-11-07

**Soundness:** 3
**Presentation:** 2
**Contribution:** 2
**Rating:** 3
**Confidence:** 3

**Summary:**

The problem and motivation is well framed and well known and the idea of parameterize the control parameters with human feedback is relevant. However, from the motivation of using LLMs to produce parametrized control to the results obtained do not "Conclusion.The experimental results demonstrate that the proposed framework is capable of training intelligent models that can fine-tune various robotic control methods across diverse environments to meet human requirements." This works looks interesting as a method for tuning controllers with natural language, but it is way far from embodied intelligence and empathy keywords that are described in the paper.

**Strengths:**

* Clear and sound motivation
* Continuous control parametrized by language

**Weaknesses:**

* LLM analysis is missing
* Parametrization is not well described. Provide the parameters used in each controller and their definition.
* Empathy is not provided to the robot. In such a case, perceived empathy, but I doubt it.
* Promising direction. While the authors talk about using the method for more complex controllers, the challenging direction is to produce more complex behaviours.

**Questions:**

How the human knows the reference control parameters?

Please provide a clear description of this: "Prompt model M to understand the relationship between control parameters and feedback"

Could you provide a better analysis of the method and not only two exemplary cases. (the appendix gives more information on how it works than the methods section.

What is the LLM architecture backbone?

**Details Of Ethics Concerns:**

However, Safety in using human input in natural language for a critical control plant should be addressed.

---

> ### Author Response · Authors · 2024-12-02
>
> Thank you for your feedback. We are currently attempting to validate this framework in more complex environments, but due to time constraints, it may be difficult to complete. We have added a supplementary introduction on empathy in the article. Below are the answers to your questions.
> Q1:
> Human do not know the reference control parameters, they just know the control result and give an opinion.
> Q2:
> In simple terms, it refers to how the parameters in the control method exert their effects and the control outcomes generated by different control parameters.
> Q3：
> We are preparing a open source environment to train different control methods. But currently we can only show these two cases.
> Q4:
> Actually, we tried different LLMs , we got the best performance in GPT.

---

### Meta-Review · Area_Chair_9ar8 · 2024-12-20

**Metareview:**

This paper proposes a framework combining large language models (LLMs) with adaptive control techniques to adjust robot control parameters based on human feedback. While the idea of leveraging LLMs for real-time adaptation in human-robot interaction is appealing, the reviewers unanimously identified critical shortcomings regarding the paper’s contributions.

The methodology is overly simplistic and lacks novelty, repurposing established techniques without introducing substantial innovations. The conceptual framing of "empathy" is vague and unsupported by rigorous definitions or evaluations, reducing its contribution to a superficial label rather than a meaningful advancement. Experimental validation is limited to basic tasks, with no strong evidence of the framework’s applicability to more complex or real-world scenarios. Furthermore, the paper fails to compare its approach against established baselines, such as traditional optimization methods, which significantly weakens its scientific rigor.

The rebuttal was brief and did not adequately address the points raised by the reviewers. The absence of substantive changes to the paper led to no changes in the reviewers' scores. Consequently, the paper does not meet the standards for acceptance at ICLR.

**Additional Comments On Reviewer Discussion:**

The reviewers raised critical concerns about the paper’s lack of novelty, methodological simplicity, and insufficient experimental rigor. They also highlighted the need for clear definitions of key concepts, such as "empathy," and the need for comparisons with existing approaches.

The authors’ rebuttal was brief and did not provide meaningful responses or new evidence to address these concerns. No additional experimental results or comparisons were provided, and the explanations offered were largely reiterations of the original manuscript. As a result, the reviewers’ concerns remained unaddressed, and there were no changes in the scores or overall assessment.

---

### Decision · Program_Chairs · 2025-01-22

Reject